# Ensemble Learning for Heterogeneous Large Language Models with Deep Parallel Collaboration

**Yichong Huang[†], Xiaocheng Feng[†‡✉], Baohang Li[†], Yang Xiang[‡], Hui Wang[‡]**
**Ting Liu[†], Bing Qin[†‡]**
[†]Harbin Institute of Technology      [‡] Peng Cheng Laboratory
{ychuang,xcfeng,baohangli,tliu,qinb}@ir.hit.edu.cn
{xiangy,wangh06}@ir.hit.edu.cn

## Abstract

Large language models (LLMs) exhibit complementary strengths in various tasks, motivating the research of LLM ensembling. However, existing work focuses on training an extra reward model or fusion model to select or combine all candidate answers, posing a great challenge to the generalization on unseen data distributions. Besides, prior methods use textual responses as communication media, ignoring the valuable information in the internal representations. In this work, we propose a training-free ensemble framework DEEPEN, fusing the informative probability distributions yielded by different LLMs at each decoding step. Unfortunately, the vocabulary discrepancy between heterogeneous LLMs directly makes averaging the distributions unfeasible due to the token misalignment. To address this challenge, DEEPEN maps the probability distribution of each model from its own probability space to a universal *relative space* based on the relative representation theory, and performs aggregation. Next, we devise a search-based inverse transformation to transform the aggregated result back to the probability space of one of the ensembling LLMs (main model), in order to determine the next token. We conduct extensive experiments on ensembles of different number of LLMs, ensembles of LLMs with different architectures, and ensembles between the LLM and the specialist model. Experimental results show that (i) DEEPEN achieves consistent improvements across six benchmarks covering subject examination, reasoning, and knowledge, (ii) a well-performing specialist model can benefit from a less effective LLM through distribution fusion, and (iii) DEEPEN has complementary strengths with other ensemble methods such as voting[1].

## 1   Introduction

With the scaling of model capacities and data volumes, generative large language models (LLMs) have shown impressive language understanding and generation abilities, shedding light for artificial general intelligence [35, 22, 13, 28]. Due to diversities of data sources, model architectures, and training recipes, LLMs have different strengths and weaknesses in various tasks and cases. Therefore, recent research has explored the ensemble of LLMs to exploit the complementary potential [15, 19].

Existing methods can be categorized into selection-based and fusion-based ensembling. Selection-based ensembling selects the best candidate answer from all individual LLMs' answers using an additionally trained reward model [15, 31, 25, 19]. Fusion-based ensembling combines all candidate answers using a trained fusion model [15]. However, these approaches inevitably face significant

---

[1]Our code is available at: https://github.com/OrangeInSouth/DeePEn
✉ means corresponding author.

challenges in generalizing to unseen data distributions and base models. Besides, prior methods enable collaboration via conveying the textual responses between LLMs while ignoring the rich information (*e.g.,* confidence and alternative answers) in the internal representations.

An ideal solution to this issue is to apply the well-established technology of prediction fusion. [36, 24, 7, 10]. For LLM ensemble, prediction fusion works at each decoding step, averaging the probability distributions from different LLMs to determine the next token. It could not only directly apply to the ensemble of any LLMs without extra parameter training, making it more general, but leverages the informative internal representations (*i.e.,* probability distributions) as communication media. Unfortunately, the vocabulary discrepancy between different LLMs makes it unfeasible to average the distributions due to token misalignment.

In this work, we tackle this key challenge by drawing upon the cross-model invariance of relative representation, which represents each token using the embedding similarities of this token to a set of anchor tokens [21]. Specifically, we propose an ensemble framework **DEEPEN** (**Dee**p **P**arallel **En**semble), enabling distribution fusion for heterogeneous LLMs. DEEPEN transforms the probability distribution from the heterogeneous probability space to a homogeneous relative space, using a matrix formed by the relative representation of all tokens. Next, DEEPEN aggregates the relative representations of all probability distributions in the relative space, coordinating the decision on the next token. Finally, the result of aggregation is transformed back to the probability space of the main model using a search-based inverse transformation to determine the next token.

We conduct extensive experiments ranging from 2-model to 9-model ensembles, covering ensembles of models with parameters ranging from 6B to 70B, ensembles of dense and sparse models, and the ensemble of LLMs with specialist models. Experimental results on six widely-used benchmarks demonstrate that compared to baselines, DEEPEN achieves consistent improvements across all benchmarks. It is also discovered that DEEPEN has complementary strengths when combined with other ensemble methods.

## 2 Theoretical Analysis

We first introduce relative representation and then illustrate the theoretical support for our method.

### 2.1 Relative Representation

Previous study discovers that despite the misalignment between latent spaces of different neural networks, the embedding similarity between samples do not change across models [21, 11, 23]. Specifically, Moschella et al. [21] propose relative representation, which represents each sample $x^{(i)}$ by the embedding similarities to a set of anchor samples $\mathbb{A}$ ($x^{(i)}$ and $\mathbb{A}$ are identically distributed):

$$\mathbf{r}_{x^{(i)}} = (cos(e_{x^{(i)}}, e_{a^{(1)}}), ..., cos(e_{x^{(i)}}, e_{a^{(|\mathbb{A}|)}})), \tag{1}$$

where $e_{(*)}$ denotes the embedding of samples, also is absolute representation.

It is empirically evidenced that relative representations possess cross-model invariance, *i.e.,* the relative representation of the same sample keeps invariant across different models, which lays the theoretical foundation for our work to fuse heterogeneous probability distributions.

### 2.2 Theoretical Support for DEEPEN

Average probability distribution has been widely evidenced to effectively improve the predictive performance in the filed of image and text [2, 10]. For generative language models, as we understand, the underlying mechanism is to interpolate different output semantics represented by the probability distributions. However, for LLM ensemble, vocabulary discrepancy isolates these output semantics in semantic spaces with different basis vectors, making the interpolation infeasible. To tackle this challenge, we aim to enable the cross-model alignment for output semantics, *i.e.,* find a transformation to map the output semantics into a universal space. To this effect, we propose to represent the output semantics with the convex combination of relative representations of all tokens where the weight is the probability assigned to the token.

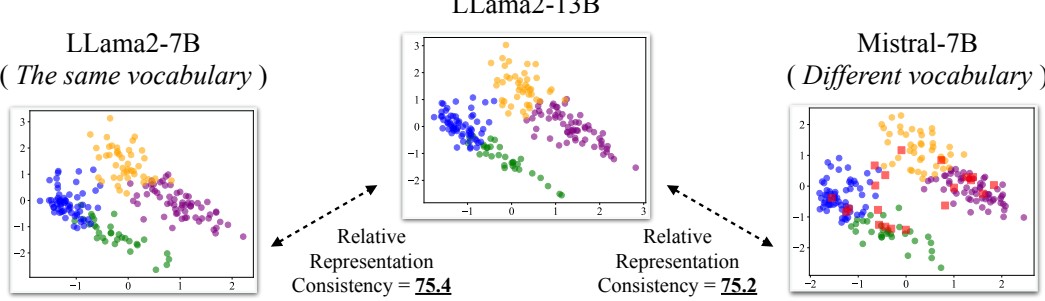

Figure 1: Visualizations for relative representations between models with the same vocabulary and between models with different vocabularies. PCA and K-means clustering are applied only for visualization. Different colors indicate different clusters of samples (word embeddings). The red block indicates the representation of tokens that only appear in Mistral's vocabulary. Relative representation consistency is obtained by calculating the cosine similarity between the relative representations of the same token in different models.

**Definition of output semantics in relative space.** Formally, given the absolute representation of the output semantics $\mathbf{p}$ and the relative representation matrix $R \in \mathbb{R}^{|V| \times |A|}$ where $V$ is the vocabulary and $A \subseteq V$ is the anchor token set. The $i$-th row of $R$ is the relative representation of word $w^{(i)}$:

$$R[i] = (cos(e_{w^{(i)}}, e_{a^{(1)}}), ..., cos(e_{w^{(i)}}, e_{a^{(|A|)}})), \tag{2}$$

and the relative representation of the output semantics $\mathbf{p}$ is defined as: $\mathbf{r} = \mathbf{p} \cdot R$.

**Model-invariance of relative representation of output semantic.** Next, we illustrate why this representation scheme could align the output semantics isolated in heterogeneous absolute spaces. First, considering two LLMs $\theta_A$ and $\theta_B$ with the same vocabulary (*e.g.,* LLaMA2-7B and LLaMA2-13B). When expressing the same output semantic, these models output the same probability distribution (*i.e.,* absolute representation) $\mathbf{p}_A$ and $\mathbf{p}_B$. Besides, they have the same (highly similar in practice) relative representation matrix due the vocabulary consistency and cross-model invariance of relative representation. Therefore, the relative representations of output semantics are also identical:

$$\mathbf{r}_A = \mathbf{p}_A \cdot R_A = \mathbf{p}_B \cdot R_B = \mathbf{r}_B. \tag{3}$$

Then, let's consider a language model $\theta_C$ with a different vocabulary (*e.g.,* Mistral). Based on the fact that different LLMs typically share mass tokens in their vocabularies (§A), the vocabulary of model $\theta_C$ is identical to adding and removing partial tokens to the vocabulary of $\theta_B$, which leads to $\mathbf{p}_B \not\cong \mathbf{p}_C$ and $R_B \not\cong R_C$. However, in our study, we discover that this change to the vocabulary has not incurred significant influence on the relative representation of the unchanged tokens (*i.e.,* the common tokens between $\theta_B$ and $\theta_C$), as shown in Fig. 1. Therefore, we make the reasonable assumption that the local change in the vocabulary could hardly influence the relative space.

## 3 Methodology

In this section, we first introduce the overall process of our ensemble framework DEEPEN and then describe the three parts of DEEPEN in detail.

### 3.1 Overview

We illustrate the process of DEEPEN in Fig. 2. Given $N$ models to ensemble, DEEPEN first constructs their transformation matrices (*i.e.,* relative representation matrices) mapping the probability distributions from the heterogeneous absolute spaces into the relative space (§3.2). At each decoding step, all models perform prediction and output $N$ probability distributions. These distributions are mapped into the relative space and aggregated (§3.3). Finally, the aggregation result is transformed back into the absolute space of the main model, in order to determine the next token (§3.4).

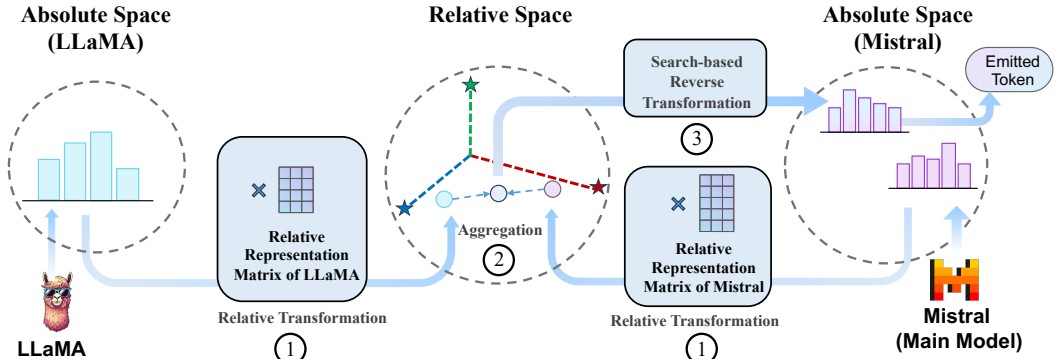

Figure 2: Overview of DEEPEN. The relative representation matrix of each LLM is directly derived by calculating the embedding similarities between each token with the anchor tokens.

## 3.2 Construction of Relative Transformation

Given $N$ models to ensemble, DEEPEN first finds out the intersection of vocabularies of all models, *i.e.,* common token set $C$, and samples a subset or uses the full set of common tokens as the anchor token set $A \subseteq C$. Next, for each model, DEEPEN calculates embedding similarities of each token to the anchor words, obtaining the relative representation matrix $R$ (as shown in Eq.2). Finally, to overcome the relative representation degeneration of outlier words, which will be introduced later, we perform normalization on the relative representation of all tokens by a softmax operation so that it becomes a probability distribution. We denote the normalized representation matrix $\hat{R}$:

$$\hat{R}[i] = softmax(R[i]). \tag{4}$$

**Anchor Selection.** The choice of anchor tokens is crucial for the relative representation capability. Previous research discovers that the capability improves as the number of anchor words increases [21]. Therefore, we employ the full set of common words between LLMs as the anchor words. It is also empirically proved that this method performs more stably on downstream tasks (§5.2).

**Normalization of relative representation matrix.** In DEEPEN, the relative representation of each token is normalized by the softmax operation to avoid the relative representation degeneration of outlier words, which are referred to as words that are far away from other words (including the anchors) and become distinguishable in relative space since for being zero vectors. The softmax operation effectively resolves this problem by making each relative representation a probabilistic distribution instead of a zero vector.

## 3.3 Aggregation in Relative Space

At each decoding step, once each model $\theta_i$ outputs the probability distribution $\mathbf{p}_i$, DEEPEN transforms $\mathbf{p}_i$ into the relative representation $\mathbf{r}_i$ using the normalized relative representation matrix: $\mathbf{r}_i = \mathbf{p}_i \cdot \hat{R}_i$, and aggregate all relative representations to obtain the aggregated relative representation:

$$\bar{\mathbf{r}} = \sum_{i=1}^{N} \alpha_i \times \mathbf{r}_i, \tag{5}$$

where $\alpha_i$ is the collaboration weight of model $\theta_i$.

**Collaboration Weight.** As our work focuses on enabling the distribution fusion of heterogeneous LLMs instead of finding the optimal collaboration weights, we follow the most common practice to uniformly aggregate the distributions ($\alpha = 1/N$, $N$ is the number of models), which is named **DEEPEN-Avg**. Besides, we also adopt a simple and effective method of deducing weights, **DEEPEN-Adapt**, which heuristically sets a larger value to the model with a better performance on the development set: $\alpha_i = s_i / \sum_j s_j$, where $s_i = Acc(\theta_i, \mathcal{D}^{dev}) - \epsilon$, $Acc(\cdot, \cdot)$ indicates the average accuracy of model $\theta_i$ on the development set, and $\epsilon$ indicates the chance level on the evaluation task. Specifically, $\epsilon = 0$ on the free-form generation tasks and $\epsilon = 1/K$ on the $K$-choice tasks.

### 3.4 Inverse Transformation of Relative Representations

To decide the next token according to the aggregated relative representation, DEEPEN aims to transform it from the relative space back to the absolute space of the main model, which is empirically selected with the best-performing model on the development set. To enable this inverse transformation, we adopt a search-based strategy, finding out the absolute representation whose relative representation is identical to the aggregated relative representation. This search problem is formulated as:

$$\overline{\mathbf{p}}_i = \underset{\mathbf{p}_i \in \mathbb{P}_i}{\arg\min} \, \ell(\mathbf{p}_i \times \hat{R}, \, \overline{\mathbf{r}}), \tag{6}$$

where $\mathbb{P}_i$ denotes the absolute space of model $\theta_i$, and $\ell(\cdot)$ is the loss function to measure the distance between relative representations. In this work, we adopt the KL-divergence due to its convergence.

This search is iteratively conducted under the guidance of the gradient of the loss in Eq.6 with respect to the absolute representation $\mathbf{p}_i$. Specifically, we initialize the start point of searching $\mathbf{p}_i^{(0)}$ with the main model's original absolute representation, and update it as:

$$\mathbf{p}_i^{(t+1)} = \mathbf{p}_i^{(t)} - \eta \times \frac{\partial \ell}{\partial \mathbf{p}_i^{(t)}}, t \in [0, T] \tag{7}$$

where $\eta$ is an important hyperparameter named the relative ensemble learning rate, and $T$ is the iterations number named relative ensemble learning steps. Finally, we use the updated absolute representation $\mathbf{p}_i^{(T)}$ to determine the emitted token.

## 4 Experiments

### 4.1 Experimental Setup

**Benchmarks.**    We mainly conduct experiments on six benchmarks, which can be categorized into:

- **Comprehensive Examination:** (1) MMLU (5-shot) [12], which covers 57 subjects that humans learn, and (2) ARC-C (0-shot) [5], collected from standardized natural science tests.
- **Reasoning Capabilities:** (1) GSM8K [6] (4-shot), which is a dataset of high quality problems at the grade school math level, and (2) PIQA [3] (0-shot), which is a commonsense reasoning dataset.
- **Knowledge Capacities:** (1) TriviaQA (5-shot) [16], collected by Trivia enthusiast authored, and (2) NQ (5-shot) [18], which is a QA corpus consists of queries issued to the Google search engine.

**Evaluation.**    For all benchmarks, we follow the test scripts of OpenCompass leaderboard. Specifically, on the multiple-choice tasks (MMLU, ARC-C, and PIQA), the option with the highest likelihood is selected to calculate the accuracy. On the free-form generation tasks (GSM8K, TriviaQA and NQ), we calculate the exact match (EM) accuracy.

**Individual models.**    As ensemble learning typically works on models with comparable performance [24, 34], we select six well-performing LLMs whose performance are closely matched: LLaMA-2-13B [29], Mistral-7B-v0.1 [13], InternLM-20B [26], Yi-6B [1], Skywork-13B-base [32], and Tigerbot-13b-base-v2 [4]. To achieve better ensemble performance, we conduct experiments on the ensemble of the top-2 models and the top-4 models for each benchmark. Besides, we also consider ensembling various number of models (§4.3) and ensembling more diverse models (§5.1).

**Hyperparameters.**    In this work, we select all of the common tokens between LLMs as the anchor tokens to build the relative spaces, *i.e.,* $A = C$ (§5.2). For the inverse transformation of relative representations, we search the optimal relative learning rate ($\eta$ in Eq. 7) from 0.05 to 0.30 with an interval of 0.05. We empirically set the number of relative ensemble learning steps $T = 5$ (§5.3).

**Comparative methods.**    We compare DEEPEN with (1) **MINED** [30, 9], which maps the probability distributions of heterogeneous LLMs to the distribution of the main model via aligning tokens in different vocabularies with edit distance, and (2) **LLM-BLENDER** [15], which comprises a reward model PAIRRANKER to score each response of LLMs and a fusion model GENFUSER to fuse

| Models | Examination | | Reasoning | | Knowledge | |
|---|---|---|---|---|---|---|
| | MMLU | ARC-C | GSM8K | PIQA | TriviaQA | NQ |
| *Individual Models* | | | | | | |
| LLaMA2-13B | 55.07 | 59.32 | 29.80 | 59.68 | 74.32 | 28.67 |
| InternLM-20B | 59.94 | 75.81 | 53.83 | 64.78 | 66.88 | 26.09 |
| Skywork-13B | 61.16 | 66.50 | 53.90 | 74.04 | 58.65 | 19.75 |
| Tigerbot-13B | 51.95 | 57.44 | 48.82 | 68.28 | 66.22 | 22.71 |
| Mistral-7B | 62.13 | 73.33 | 47.50 | 65.61 | 73.18 | 27.62 |
| Yi-6B | 63.25 | 73.33 | 37.91 | 76.15 | 59.02 | 18.98 |
| *Top-2 Ensemble* | | | | | | |
| LLM-BLENDER | 63.85 (+0.60) | 75.73 (- 0.08) | 54.89 (+0.99) | 78.31 (+2.16) | 74.10 (- 0.22) | 28.61 (- 0.06) |
| MINED | 65.04 (+1.79) | 77.35 (+1.54) | 18.50 (-35.40) | 78.98 (+2.83) | 72.30 (- 2.02) | 28.45 (- 0.22) |
| **DEEPEN-Avg** | 64.68 (+1.43) | 77.52 (+1.71) | 55.42 (+1.52) | 78.87 (+2.72) | 75.90 (+1.58) | 30.17 (+1.50) |
| **DEEPEN-Adapt** | 65.01 (+1.76) | 77.52 (+1.71) | 55.65 (+1.75) | 79.37 (+3.22) | 76.08 (+1.76) | 30.69 (+2.02) |
| *Top-4 Ensemble* | | | | | | |
| LLM-BLENDER | 61.44 (- 1.81) | 71.03 (- 4.78) | 43.37 (-10.53) | 71.16 (- 4.99) | 67.87 (- 6.45) | 24.18 (- 4.49) |
| VOTING | 64.88 (+1.63) | 78.41 (+2.60) | 63.15 (+9.25) | 76.82 (+0.67) | — | — |
| MBR | — | — | 62.09 (+8.26) | — | 74.32 (+0.00) | 30.28 (+1.61) |
| MINED | 65.61 (+2.36) | 78.68 (+2.87) | 56.56 (+2.66) | 77.87 (+1.72) | 71.62 (- 2.70) | 29.50 (+0.83) |
| **DEEPEN-Avg** | 65.09 (+1.84) | 78.70 (+2.89) | 56.18 (+2.28) | 77.15 (+1.00) | 75.74 (+1.42) | 31.55 (+2.88) |
| **DEEPEN-Adapt** | 65.25 (+2.00) | 79.15 (+3.34) | 56.25 (+2.35) | 78.59 (+2.44) | 75.76 (+1.44) | 31.77 (+3.10) |
| +VOTING/MBR | 65.40 (+2.15) | 79.44 (+3.63) | 65.25 (+11.35) | 77.37 (+1.22) | 75.65 (+1.33) | 32.11 (+3.44) |

Table 1: Main results. The best individual model is highlighted in red , and the best ensemble method is highlighted in green, except for the results of the combined method (i.e., the last row). The top-4 models on each benchmark are underlined. '—' indicates that the method does not apply to the task.

candidate responses. In this work, we we only adopt the PAIRRANKER since GENFUSER suffers from serious over-generation under our training-free setting. In the ensemble of more than two models, we introduce two additional ensemble methods: (3) **VOTING**, which selects the choice favored by most models on the tasks with outputs limited to a fixed set, and (4) **MBR** [8, 17], which selects the answer with the highest textual similarity to other candidate answers. The implementation details of baselines are illustrated in §B.

## 4.2 Main Results

The main results are shown in Tab. 1, from which we have drawn the following observations:

**(1) DEEPEN achieves consistent improvements over the individual models.** These results prove that our DEEPEN successfully enables collaboration between heterogeneous LLMs via aggregating their probability distributions in the relative space. Specifically, DEEPEN-Avg achieves improvements of +1.43(MMLU)∼+2.72(PIQA) on the ensemble of top-2 models, and +1.00(PIQA)∼+2.89(ARC-C) on the ensemble of top-4 model. DEEPEN-Adapt gains improvements of +1.44(TriviaQA)∼+3.34(ARC-C) on the ensemble of top-4 models.

**(2) DEEPEN shows better stability than baselines.** As shown, LLM-BLENDER struggles to achieve improvements under the training-free setting. MINED shows unstable performance across different benchmarks. For example, MINED leads to performance drops of -35.40 on the GSM8K benchmark under the top-2 models ensemble setting and -2.70 on the TriviaQA, indicating the limitation of using textual similarity to align tokens in heterogeneous vocabularies. Through case studies, it is revealed that this method of aligning tokens with edit distance disturbs the decoding and produces incomplete words (demonstrated in §7). Instead, DEEPEN-Avg achieves consistent improvements and surpasses all baselines in 7/12 settings.

**(3) DEEPEN has complementary strengths with other ensemble methods.** VOTING achieves a significant improvement on the mathematical reasoning GSM8K, showing the effectiveness of

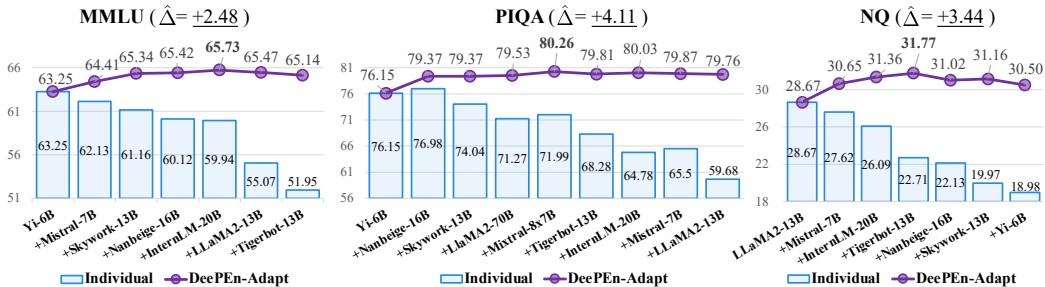

Figure 3: Test set results of ensemble learning on various number of models. Individual models are arranged in descending order of their performance on the development set, and sequentially incorporated into the ensemble. $\hat{\Delta}$ indicates the largest improvement achieved by DEEPEN.

| Model | GSM8K | PIQA |
|---|---|---|
| LLaMA2-70B | 63.84 | 71.27 |
| Mixtral-8×7B | 65.73 | 71.88 |
| LLM-BLENDER | 64.52 (-1.21) | 74.54 (+2.66) |
| MINED | 67.10 (+1.37) | 75.65 (+3.77) |
| **DEEPEN** | 67.33 (+1.60) | 75.10 (+3.22) |

Table 2: Ensemble learning of the *dense* large language model LLaMA2-70B and the *sparse* MoE model Mixtral-8×7B.

| Model | En→De | De→En | En→Ro | Ro→En |
|---|---|---|---|---|
| LLaMA2-13B | 30.60 | 42.27 | 30.83 | 39.99 |
| NLLB-600M | 32.30 | 41.49 | 31.91 | 42.39 |
| LLM-BLENDER | 33.26 (+0.96) | 43.28 (+1.01) | 33.17 (+1.26) | 41.99 ((-0.40)) |
| MINED | 27.12 (-5.18) | 36.83 (-5.44) | 29.91 (-2.00) | 34.39 ((-8.00)) |
| **DEEPEN** | 33.34 (+1.04) | 43.70 (+1.43) | 32.95 (+1.04) | 42.84 (+ 0.45) |

Table 3: Ensemble learning of the *generalist* model LLaMA2 and the *specialist* translator model NLLB on the translation benchmark Flores-200.

reasoning with multiple paths. To evidence the complementary strength of DEEPEN with VOTING, we combine both methods. On the TriviaQA and NQ, VOTING is replaced with MBR. As shown that the combination of both methods gains a further improvement over VOTING (63.15→65.25).

**(4) Collaboration with more worse-performing LLMs is a double-edged sword.** The ensemble performance of DEEPEN-Avg with top-4 models surpasses that with top-2 models on 4 benchmarks, but falls short on 2 benchmarks. This is reasonable because incorporating the 3rd and 4th ranked LLMs enhances complementary strengths but also causes the interference with the top-2 models.

### 4.3 Results on Different Numbers of Models

Next, we illustrate the effectiveness of DEEPEN on the ensemble of more models on the MMLU, PIQA, and NQ. We add Nanbeige-16B into the ensemble on all three benchmarks, and add LLaMA2-70B and Mixtral-8×7B on the PIQA due to their comparable performance. As illustrated in Fig. 3, the ensemble performance increases first and then decreases with the joining of more models in descending order of performance. And the ensemble performance peaks in the top-4 or top-5 models across three benchmarks.

## 5 Analysis

To deeply understand DEEPEN, we first evaluate its performance on the ensemble learning of model sets with diverse architectures, abilities, and performance gaps. Next, we conduct a series of analyses on the reverse transformation process of relative representations.

### 5.1 Results of Ensembling Diverse Models

**Ensemble of the dense model and the sparse model.** We first evaluate our method on the ensemble learning of the dense model and the sparse MoE model on the challenge reasoning tasks. Specifically, we use the widely-used large-scale dense model LLaMA2-70B [29] and the popular sparse MoE model Mixtral-8×7B [14] as the base models. As the results shown in Tab. 2, our DEEPEN achieves improvements of +1.60 and +3.22 on the GSM8K and PIQA datasets, even though the base models have achieved a high level of performance.

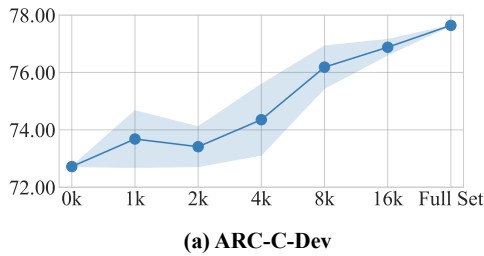

(a) ARC-C-Dev

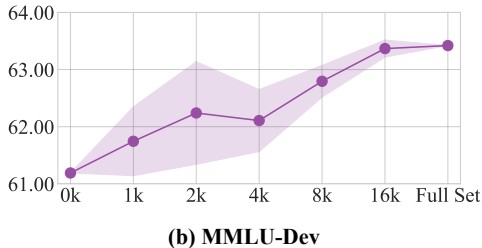

(b) MMLU-Dev

Figure 4: Effect of the number of anchor words. The x-axis indicates the number of anchor words randomly sampled from the common words for 4 times.

| Methods | MMLU-Dev | | TriviaQA-Dev | |
|---|---|---|---|---|
| | ACC | Δ | ACC | Δ |
| **Baseline** | 61.19 | – | 72.74 | – |
| **DEEPEN** | 63.61 | +2.42 | 74.79 | +2.05 |
| w/o. **Rel-Norm** | 60.73 | -0.46 | 72.95 | +0.21 |

Table 4: Ablation study of normalization on the relative representation matrix to the ensembling performance on the development sets. **Baseline** refers to as the best single model on each benchmark. **DEEPEN** refers the performance of ensembling top-2 models in the benchmark.

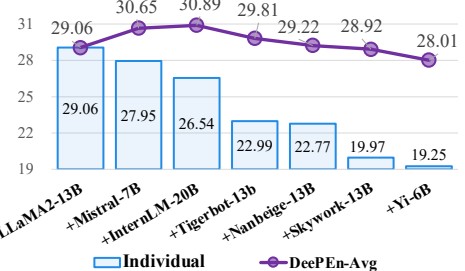

Figure 5: 2-model ensemble of the top-1 model (LLaMA2-13B) with different models on the NQ benchmark, respectively.

**Ensemble of the generalist model and the specialist model.** To investigate the effectiveness of DEEPEN on the ensemble of the generalist model and the specialist model for the specific task, we conduct experiments on the machine translation task using the ensemble of the large language model LLaMA2 and the machine translation model NLLB [27], which is a well-known open-source multilingual translator. We adopt the widely-used machine translation benchmark Flores-200[2]. As the results in Tab. 3 illustrated, DEEPEN achieves better translation performance leveraging the diverse translation knowledge in the generalist LLM and the specialist translator.

**Ensemble of models with different performance gaps.** To assess the stability of DEEPEN regarding to the performance gap of base models, we conduct an experiment on the ensemble of model pairs with increasing performance gaps. As the result demonstrated in Tab. 5, the performance of ensemble learning between a well-performing model (the rank-first model)with a worse-performing model could achieve improvements or slightly lag behind the well-performing model.

### 5.2 Analysis on Relative Transformation

**Effect of anchor selection.** We demonstrate the impact of different numbers of anchor words through experiments with the top-2 ensemble models on the MMLU and ARC-C datasets. As shown in Fig. 4, an increased number of anchor words can improve performance for LLMs in downstream tasks, and selecting the full set of common words as anchors provides better performance.

**Effect of normalization on relative representation matrix.** To demonstrate the importance of normalization on the relative representation matrix to the ensemble performance (§3.2), we conduct an ablation analysis. The result is shown in Tab. 4, the ensemble struggles to achieve improvements due to the ineffective representation of outlier words, *i.e.,* words distant to other words. The proportion of outlier words can be derived from the distribution of distance to nearest neighbor words, which is illustrated in Fig. 8. As illustrated, a remarkable proportion ($> 30\%$) of words are distant from other words, *i.e.,* cosine similarity to its nearest neighbor word is less than 0.3. Through the normalization operation, the output semantics that intend to emit outlier words could be prevented from becoming zero vectors by relative transformation.

---

[2]https://github.com/facebookresearch/flores

## 5.3 Analysis of Reverse Transformation

To better understand the reverse transformation process (§3.4) transforming the relative representation back to the absolute space of the main model, we further analyze each component of this process.

**Analysis of relative ensemble learning rates.** As shown in Tab. 5, the performance of DEEPEN is sensitive to the value of relative ensemble learning rate ($\eta$), which is abbreviated by RELR. This observation motivates us to measure the generality of this hyperparameter. Specifically, we illustrate the cross-distribution performance of the searched optimal value of $\eta$ in Tab. 9. As observed, the optimal value of RELR varies across different datasets, which suggests that the inverse transformation from relative space to absolute space requires adaptive mapping schemes.

**Effect of iteration steps in relative ensemble learning.** To give a deep view of the dynamics of the inverse transformation in DEEPEN, we report the performance change along with different numbers of relative ensemble learning steps ($T$). Besides, the dynamics of loss of relative ensemble learning ($\eta$ in Eq. 6)is also reported. As shown in Fig. 9, on the one hand, more steps of relative ensemble learning significantly lead to lower losses. However, the loss is hard to reach zero, *i.e.,* under-

| RELR ($\eta$) | 0.05 | 0.10 | 0.15 | 0.20 | 0.25 | 0.30 |
|---|---|---|---|---|---|---|
| **MMLU** | +2.42 | +1.57 | +1.77 | +1.96 | +1.31 | +1.31 |
| **TriviaQA** | +1.31 | +2.05 | +1.63 | +1.94 | +1.82 | +1.26 |

Table 5: Sensitivity analysis of relative ensemble learning rate (**RELR**). We report the improvements of ensembling top-2 models over the best individual models.

fitting. On the other hand, increasing the number of steps of relative ensemble learning will cause the performance to increase first and then decrease. The reason behind the performance drop could be that in the early stage of optimization, the focus of optimization is on updating the high-probability tokens. In the later stage of optimization, since the probabilities of all words will be adjusted equally, the high-probability tokens will be interfered with the high-probability ones, thus affecting the performance. Therefore, it is recommended to set a modest value of step number (*e.g.,* $T = 5$).

## 6 Related Work

**Selection-based ensemble.** *Rerank* is an intuitive solution to utilize multi-model strengths. Jiang et al. [15] take the first step towards LLM ensemble, training a reward model PAIRRANKER for pairwise comparison on candidate outputs. To overcome the huge computation costs of multi-LLM inference, several works have explored to train a *router* to predict the best-performing model out of a fixed set of LLMs for the given input [31, 25, 19].

**Fusion-based ensemble.** Towards a synergy between LLMs, Jiang et al. [15] propose GENFUSER, trained to combine multiple candidate answers. Different from these training-dependent ensemble methods which pose a great challenge to the generalizability of the reward model or fusion model, our DEEPEN is completely training-free, making it more general. Similar to our method, MINED also aims to tackle the vocabulary discrepancy via aligning the tokens in different vocabularies based on edit distance [30, 9]. Unfortunately, this textual similarity-based method exhibits unstable performance and produces abnormal text for LLM ensemble (Tab. 7).

There are several contemporaneous works related to our work. Xu et al. [33] propose EVA to tackle vocabulary discrepancy by learning token alignment between different vocabularies with the assistance of overlapping tokens. Our DEEPEN eliminates this training process via directly aligning tokens with the relative representation (more discussion is illustrated in §B). Mavromatis et al. [20] explore adaptive collaboration weights at test time by harnessing the perplexity on the input prompt. We emphasize that this work is complementary to our work.

## 7 Conclusion

In this work, we propose a training-free LLM ensembling framework DEEPEN, which addresses the vocabulary discrepancy when fusing the probability distributions of heterogeneous LLMs. Experimental results on six widely-used benchmarks demonstrate that DEEPEN exhibits more stable

performance than baseline methods and has complementary strengths with other ensemble methods such as VOTING. We believe our work can inspire further research on the LLMs collaboration, model reuse, and knowledge distillation. In the future, we aim to explore more effective adaptive collaboration schemes to leverage the complementary strengths between different LLMs.

## Acknowledgements

Xiaocheng Feng is the corresponding author of this work. We thank the anonymous reviewers for their insightful comments. This work was supported by the National Natural Science Foundation of China (NSFC) (U22B2059, grant62276078), the Key R&D Program of Heilongjiang via grant 2022ZX01A32, the International Cooperation Project of PCL, PCL2022D01 and the Fundamental Research Funds for the Central Universities (Grant No.HIT.OCEF.2023018).

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

| | Mistral-7B | InternLM-20B | Skywork-13B | LLaMA2-13B | Yi-6B | Tigerbot-13B |
|---|---|---|---|---|---|---|
| **Mistral-7B** | 32,000 | 26,759 | 24,983 | 24,184 | 24,360 | 25,121 |
| **InternLM-20B** | 26,759 | 103,168 | 41,204 | 22,566 | 50,362 | 44,885 |
| **Skywork-13B** | 24,983 | 41,204 | 65,519 | 32,000 | 33,646 | 49,693 |
| **LLaMA2-13B** | 24,184 | 22,566 | 32,000 | 32,000 | 20,301 | 32,000 |
| **Yi-6B** | 24,360 | 50,362 | 33,646 | 20,301 | 64,000 | 39,360 |
| **Tigerbot-13B** | 25,121 | 44,885 | 49,693 | 32,000 | 39,360 | 60,515 |

Figure 6: Statistics of common words across different vocabularies.

# A    Statistics of Common Tokens across different LLMs

We count the number of common tokens shared among different LLM vocabularies and present the results in Fig. 6. It is observed that a large number of common words (>20k) exist across the different vocabularies. We also count the number of common tokens in all six LLMs and find that there are a total of 18k common tokens, enabling DEEPEN to be applied to the ensemble learning of a large number of models.

# B    Details of Baselines

**LLM-BLENDER.**    (1) the selection-based ensemble method **PAIRRANKER** Jiang et al. [15], which is a reward model to score each response of LLMs and (2) the fusion-based ensemble method **GENFUSER** Jiang et al. [15], which is a generative model to fuse multiple candidate responses. Both models are trained on the constructed instruction tuning dataset MixInstruct. In our experiments, as GENFUSER struggles to generate responses following the expected format, we only adopt PAIRRANKER.

**VOTING.**    For tasks with outputs limited to a fixed set (*i.e.,* MMLU, ARC-C, PIQA, GSM8K benchmarks), we adopt the VOTING method on the ensemble learning of more than 2 models. Concretely, we count each candidate answer's occurrences and select the most frequent as the final output. In the event of a tie, the main model's answer is used as the final output.

**MBR.**    For generation tasks, we implement the MBR [8, 17] method, which selects the answer with the highest lexical similarity to other candidate answers. To measure this similarity, we experimented with the edit distance and chrF[3] metrics, ultimately choosing chrF due to its superior performance.

**MINED.**    To bridge the gap between different vocabularies in LLM ensemble, MINED apply the Minimum Edit Distance (MinED) approach to align tokens across different vocabularies, *e.g.,* "get" to "gets". However, this textual similarity-based mapping method could disturb the text generation process and produce incomplete words.

**EVA.**    Recently, Xu et al. [33] propose EVA to tackle the vocabulary discrepancy by learning mappings between the vocabularies of different LLMs with the assistance of overlapping tokens. We have tried to re-implement their method with the released code. However, we encounter a technical problem in that EVA only supports the ensemble learning between LLMs with the same embedding dimension. This is caused by the limitation of tool of vecmap[4], which is used to learn the token alignment.

**Error Analysis on Generation Process of MinED**

| | | |
|---|---|---|
| Question | Which Lloyd Webber musical premiered in the US on 10th December 1993? | In which American state is the Isabella Stewart Gardner Museum? |
| Golden Answer | Sunset Boulevard | Massachusetts |
| MinED Answer | unmasked | assachusetts |
| DeePEn Answer | Sunset Boulevard | Massachusetts |
| Top-10 tokens output by Main model | ['S', 'Ph', 'The', 'Wh', 'C', 'J', 'As', 'Jose', 'B', 'Ste'] | ['M', 'B', 'The', 'Is', 'MA', 'In', 'New', '_Massachusetts', 'N', 'Connect'] |
| Top-10 tokens output by Assistant Model | ['Sun', 'Wh', 'The', 'Ph', 'C', 'S', 'Sch', 'J', 'Ev', 'As'] | ['Mass', 'B', 'M', 'MA', 'New', '_Massachusetts', 'The', 'In', 'Conne', '<0x0A>'] |
| Mapped Top-10 tokens of Assistant Model | ['un', 'Wh', 'The', 'Ph', 'C', 'S', 'Sch', 'J', '_v', 'As'] | ['ass', 'B', 'M', 'MA', 'New', '_Massachusetts', 'The', 'In', 'Conne', '<0x0A>'] |
| Averaged Top-10 Tokens | ['un', 'S', 'The', 'J', '_Joseph', 'Ph', 'Wh', 'C', 'As', 'Jose'] | ['ass', 'M', 'B', 'C', 'MA', 'The', 'New', '_Massachusetts', 'Is', 'In'] |
| | **Disturb Decoding** | **Produce Incomplete Words** |

Figure 7: Analysis of the generation process of MINED. To illustrate the problematic generation process of MINED, we list the top-10 high-probability tokens in the probability distribution of the assistant model and their aligned token.

| Models | MMLU-Dev | | ARC-C-Dev | |
|---|---|---|---|---|
| | INDIV | DEEPEN | INDIV | DEEPEN |
| Yi-6B | 61.19 | **63.61** (+2.42) | 72.72 | **77.55** (+4.83) |
| Mistral-7B | 60.80 | **64.46** (+3.66) | 73.88 | **77.73** (+3.85) |

Table 6: Performance of DEEPEN with choosing different main models on the development sets. INDIV refers to as individual models. The result of DeePEn indicates the performance of using the model of this row as the main model.

## C   Additional Experiments

### C.1   Choice of main model.

In the process of inverse transformation, DEEPEN maps the relative aggregated representation to the absolute space of the main model. Ideally, we expected the results of inverse transformation to keep invariant with the choice of the main model. However, this objective is hard to achieve due to the underfitting observed in the search process. Therefore, we illustrate the performance variance of choosing different main models in Tab. 6. As the results shown on ARC-C, changing the main model from the first-ranked Mistral-7B to the second-rank Yi-6B, the ensemble performance is decreased slightly from 77.73 to 77.55. Interestingly, changing the main model from the rank-1 Yi-6B to the rank-2 Mistral-7B on **MMLU**, the performance is actually improved from 63.63 to 64.46, which indicates that Mistral-7B benefits more than Yi-6B from collaboration. Even so, choosing different main models does not significantly affects the ensemble performance.

### C.2   Comparison to Vanilla Prediction Average

To compare our DEEPEN with vanilla prediction average, we conduct an experiment for ensembling two LLMs with the same vocabulary and comparable performance on MMLU, *i.e.,* LLaMA2-7B and LLaMA1-13B. As shown in Tab. 7, the performance of DEEPEN is comparable, even better than, that of the vanilla prediction average. Theoretically, the performance of the vanilla prediction average is the performance upper-bound of DEEPEN. The reason that DEEPEN could excel over the vanilla one on MMLU is the under-fitting in the inverse transformation process, which leads to the weights to aggregate the output semantics of different models not being a uniform distribution (*i.e.,* $(0.5, 0.5)$).

---

[3]`https://github.com/mjpost/sacrebleu`
[4]`https://github.com/artetxem/vecmap`

| Models | MMLU-Dev | | | MMLU-Test | | |
|---|---|---|---|---|---|---|
| | INDIV | VANIL | DEEPEN | INDIV | VANIL | DEEPEN |
| LLaMA1-13B | 43.26 | 45.48 | 44.37 | 43.70 | 45.01 | 44.22 |
| LLaMA2-7B | 42.28 | | **45.94** | 42.99 | | **45.31** |

Table 7: Comparison to vanilla prediction average (VANIL) on the ensemble of LLMs with the same vocabulary.

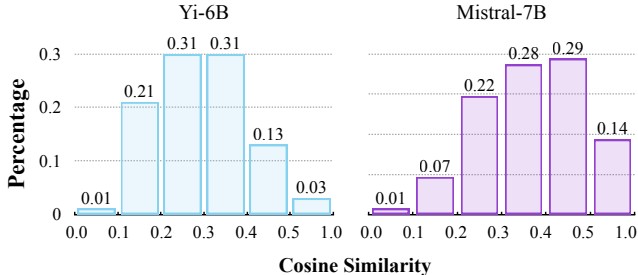

Figure 8: Distance distribution to nearest neighbor words. The distance is measured by calculating the cosine similarity between words.

For example, in Tab. 7, the weights for LLaMA1 and LLaMA2 could be $(0.6, 0.4)$, where the weight of the main model is larger than the other model.

### C.3 Latency Analysis

To accomplish the fusion of heterogeneous distributions, DEEPEN first maps the distributions into the relative space and adopts the search-based inverse transformation to map the aggregated relative representation back to the main model's probability distribution, which incurs an extra latency. This latency is mainly caused by the inverse transformation process, which requires $T$-round search. To demonstrate this latency, we report the token-level inference latency of ensembling two LLMs (Mixtral-8×7b and LLaMA2-70B). This experiment is conducted on 8 A100 GPUs. All of our experiments can be re-implemented on 8 A100 GPUs. As shown in Tab. 8, DEEPEN causes +17% token-level inference latency. However, in practice, this latency could be greatly decreased since all individual models intend to emit the same token in 90% decoding steps. In these steps, we could skip the fusion process and use the consistently agreed token as the next token. In total, DEEPEN actually incurs less than 2% sentence-level inference latency.

| | Baseline | $T = 1$ | $T = 3$ | $T = 5$ | $T = 10$ |
|---|---|---|---|---|---|
| **Inference Latency** | 0.19s | 0.20s | 0.21s | 0.22s | 0.24s |
| **Relative Change** | 0% | +7% | +11% | +17% | +29% |

Table 8: Inference Latency of DEEPEN with different search steps $T$.

## D Limitations

As illustrated in Tab. 1, collaboration with more LLMs can sometimes lead to a performance drop caused by interference from lower-performing models. This issue limits the ensemble performance of our current method, even though we have explored setting different collaboration weights for each model on each benchmark (DEEPEN-Adapt). An ideal solution would be to set adaptive collaboration weights at the sample level, or even the token level, for each LLM, which remains a significant

|          | Baseline | TrivaQA | NQ    | ARC-C | MMLU  |
|----------|----------|---------|-------|-------|-------|
| TriviaQA | 73.42    | 75.9    | 75.41 | 75.56 | 75.44 |
| NQ       | 29.11    | 30.55   | 30.65 | 30.42 | 30.69 |
| ARC-C    | 60.29    | 69.32   | 72.31 | 74.19 | 73.76 |
| MMLU     | 54.06    | 59.97   | 61.04 | 61.94 | 61.42 |

Table 9: Cross-distribution validation of relative ensemble learning rate ($\eta$). We report the performance of ensembling LLaMA2-13B and Mistral-7B. Each row indicates the test set used to evaluate performance. Each column indicates the development set used to search the optimal value of $\eta$.

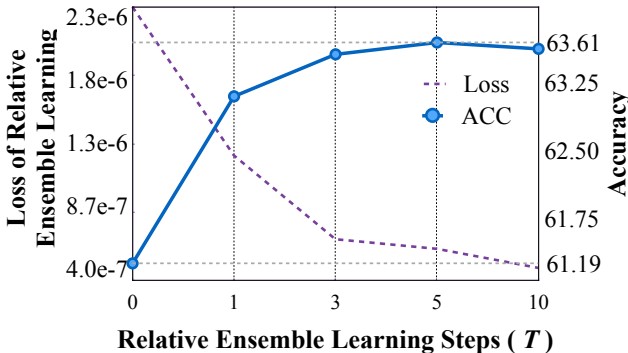

Figure 9: Effect of different number of relative ensemble learning steps.

challenge. Despite this, our work represents an important step towards the distribution fusion of LLMs.

