# OpenReview forum: "Ensemble Learning for Heterogeneous Large Language Models with Deep Parallel Collaboration"
_NeurIPS.cc/2024/Conference — NeurIPS 2024 spotlight_

### Official Review · Reviewer_JJTA · 2024-07-08

**Soundness:** 3
**Presentation:** 3
**Contribution:** 3
**Rating:** 7
**Confidence:** 3

**Summary:**

The authors propose to use relative representations to unify the tokenization across different models for an effective ensemble. They first transform the prediction of each model to a relative space. Then, after averaging the relative predictions, they perform a gradient-based optimization to find the averaged prediction in the original space.

**Strengths:**

1. The authors propose an interesting idea for dealing with different tokenization.
2. The authors provide detailed analysis such as learning rate and the number of steps for the search step.

**Weaknesses:**

1. The proposed method requires gradient updates to project back to the original space at every generation step. This can cause extra overhead.
2. The performance largely depend on the number of common anchor words between different LLMs (Figure 4). As a result, the method may be not effective when the domain of pretrained LLMs are very different.

**Questions:**

1. Have you tried other normalization methods other than softmax?

**Limitations:**

Discussed in Appendix.

---

> ### Author Rebuttal · Authors · 2024-08-06
>
> Thanks for your insightful comments, which greatly help us improve our paper. We are glad to have this discussion to address your concerns.
>
> &nbsp;
>
> **Concern-1: The proposed method requires gradient updates to project back to the original space at every generation step. This can cause extra overhead.**
>
> Thanks for your insightful comment. We understand your concern considering the overhead of the reverse transformation process from the relative space to the absolute one. To address this concern, we quantitatively analyzed the overhead of this process in Appendix C.3.
>
> **Overall, DeePEn actually incurs $\leq $2% inference latency in practice. This conclusion also applies to the overhead.**
>
> &nbsp;
>
> **Concern-2: As a result, the method may not be effective when the domain of pretrained LLMs are very different.**
>
> Thanks for your insightful comment. We understand your concern regarding the number of common words between LLMs with quite different vocabularies. To address this concern, we analyzed the number of common words across different LLMs in Appendix. A (Fig. 6). Through case studies, we found that a large number of common words (>20k) exist across the different vocabularies since **LLMs (including LLMs tailored for specific domains) usually contain a lot of common English basic tokens.** This fact provides a solid guarantee for our DeePEn to work effectively.
>
> &nbsp;
>
> **Question-1: other normalization methods other than softmax?**
>
> Thanks for your constructive suggestion. We follow your advice to experiment with an alternative normalization method of Softmax. Concretely, we replace softmax with Min-Max Scaling:
>
> $ x{\prime} = \frac{x - x_{min}}{x_{max} - x_{min}} $
>
> The result shows that the normalization with softmax is better than the one with Min-Max Scaling:
>
> | Methods                                   | MMLU-Dev  | TriviaQA-Dev |
> | ----------------------------------------- | --------- | ------------ |
> | Baseline                                  | 61.19     | 72.74        |
> | DeePEn w/o. Normalization                 | 60.73     | 72.95        |
> | DeePEn w. Normalization (Softmax)         | **63.61** | **74.79**    |
> | DeePEn w. Normalization (Min-Max Scaling) | 60.67     | 73.55        |
>
> We argue that Softmax pays more attention to the nearest neighbor words when modeling the structural relation (relative representation) than Min-Max Scaling, which could be more beneficial.
>
> We will add this analysis to our paper.

---

> > ### Comment · Reviewer_JJTA · 2024-08-14
> >
> > Thanks for the detailed response! I will maintain my positive stance towards the paper.

---

### Official Review · Reviewer_Q9GV · 2024-07-11

**Soundness:** 3
**Presentation:** 3
**Contribution:** 3
**Rating:** 6
**Confidence:** 4

**Summary:**

The paper introduces DEEPEN, a novel training-free ensemble framework designed to leverage the complementary strengths of various large language models (LLMs). The key innovation of DEEPEN lies in its ability to fuse informative probability distributions from different LLMs at each decoding step, addressing the challenge of vocabulary discrepancies between heterogeneous LLMs.

DEEPEN operates by transforming the probability distributions from each model's vocabulary space into a shared "relative representation" space based on the relative representation theory, which uses embedding similarities to a set of anchor tokens. The aggregated relative representations are then mapped back to the vocabulary space of one LLM (the main model) to determine the generated token.

**Strengths:**

1) The paper introduces DEEPEN, a novel ensemble learning framework that enables collaboration among heterogeneous large language models without the need for additional training. A significant contribution is the solution to the vocabulary mismatch problem between different LLMs, allowing for more effective fusion of their outputs.
2) The paper includes extensive experiments across six benchmarks, demonstrating DEEPEN's consistent improvements in performance and its complementary strengths with other ensemble methods.
3) DEEPEN demonstrates better stability compared to baseline methods, which struggle with generalization to unseen data distributions.
4) The framework is open to further extensions and improvements, such as the development of more sophisticated methods for determining collaboration weights.

**Weaknesses:**

1) The paper does not provide a detailed analysis of how well DEEPEN generalizes to unseen data or across different types of tasks beyond the evaluated benchmarks.
2) The method's performance is sensitive to the choice of hyperparameters, such as the relative ensemble learning rate. Finding optimal hyperparameters may require additional tuning and may not be straightforward.
3) The performance of DEEPEN relies heavily on the selection of anchor words. The paper does not thoroughly explore the impact of different anchor word selection strategies on the ensemble performance.

**Questions:**

See weaknesses.

**Limitations:**

See weaknesses.

---

> ### Author Rebuttal · Authors · 2024-08-06
>
> Thank you for your insightful comments, which have greatly helped us improve our paper. We appreciate the opportunity to discuss and address your concerns.
>
> &nbsp;
>
> **Question-1: How well does DeePEn generalize to unseen data or across different types of tasks beyond the evaluated benchmarks?**
>
> We understand your concern regarding the generalizability of DeePEn, which relies on a development set for each task to find the optimal relative ensemble learning rate.
>
> Actually, **we analyzed the generalizability of DeePEn through *cross-distribution validation* and found that DeePEn generalizes well, which is shown in Tab.9 of the paper.**
>
> Concretely, we tested the performance of DeePEn on task $A$ with the hyperparameter found with the development set of task $B$. We totally consider 4 tasks: 2 generative knowledge QA tasks (NQ and TriviaQA), 2 multi-choice human examination tasks (MMLU and ARC-C).
>
> The results show that the performance improvement using a different-task development set ($A \neq B$) is **88%** of the improvement when using a same-task development set  ($A = B$). Moreover, using a similar-task development set, the performance improvement is **95%** of that achieved using a same-task development set.
>
> Thanks for your important feedback. We will demonstrate this important analysis more clearly by modifying Tab.9 and placing it in the main document instead of the appendix.
>
> &nbsp;
>
> **Question-2: Finding optimal hyperparameters may require additional tuning.**
>
> We acknowledge that our current method requires tuning to find the optimal relative ensemble learning rate for effective inverse mapping from the relative space to the absolute space. As the first attempt to address this challenge, we have achieved significant success. In the future, we plan to explore methods to achieve this inverse mapping without the need for extensive tuning.
>
> &nbsp;
>
> **Question-3: What impact do different anchor word selection strategies have on the ensemble performance?**
>
> Thanks for your constructive suggestion. We had ever devised an anchor selection algorithm **AS-MRRC**  (Anchor Selection with Maximum Relative Representation Consistency). AS-MRRC aims to **infer the optimal anchor words** via maximizing the relative representation similarity of common words across different models:
>
> $A^* = \mathop{argmax}\limits_{A \in C} \ \mathop{\mathbb{E}}\limits_{i\in C} \ cos(\hat{R}_1(i|A), \hat{R}_2(i|A)),$
>
> where $C$ is the common word set between different LLMs, $\hat{R}_1(i|A)$ and $\hat{R}_2(i|A)$ are the relative representations of word $i$ in different models using $A$ as anchor words, and $cos(,)$ refers to the cosine similarity function.
>
> We compared AS-MRRC with the random selection of anchor words and the method of using all common words as anchors:
>
> | Method                   | Number of Anchors | Performance (ARC-C-Dev) |
> | ------------------------ | :---------------: | :---------------------: |
> | Random Selection         |        16k        |          76.88          |
> | Full Set of Common Words |        24k        |        **77.64**        |
> | AS-MRRC                  |        13k        |          77.43          |
>
> As the result shows, our AS-MRRC outperforms random selection using fewer anchors while underperforming the method of using the full set of common words. **Therefore, we decided not to report this thankless method of AS-MRRC**.
>
> In the future version, we will follow your advice and include this trial to help readers better understand the impact of different anchor word selection strategies. We are also going to explore anchor word selection strategies more thoroughly.

---

### Official Review · Reviewer_c5c3 · 2024-07-12

**Soundness:** 3
**Presentation:** 3
**Contribution:** 3
**Rating:** 6
**Confidence:** 4

**Summary:**

The paper introduces a method that maps output distributions of different LLMs to and from a universal relative space to aggregate them, based on which the next token is determined.

**Strengths:**

- The paper is structured well and written clearly.
- It proposes a novel method of ensembling the heterogeneous output distributions of LLMs.
- The method is mathematically well-motivated.
- It shows good empirical performance and ablation studies give valuable insights.

**Weaknesses:**

- **Presentation**: Although the paper is well-structured, some passages could be condensed. For instance, the paragraphs "Anchor Selection" and "Normalization of relative representation matrix" simply repeat what has already been stated in the general paragraph above. Also, sections 3.3 and 3.5 both consider the aggregation of relative representations, currently separated by section 3.4 which considers the inverse transformation. Combining those conceptually the same topics would improve the reading flow.
- **Evaluation**: Although the paper provides numerous ablation studies, the method is only compared against the baselines in the main experiments (Tab. 1). Here, the method outperforms the baseline in (only) 7/12 settings. Also comparing to the baselines in the other experiments (see questions to Fig. 3. and Tab. 2-3 below) would provide further insights into the performance of the method.

**Questions:**

- Figure 1: what do the three colors (blue, yellow, purple) indicate?
- Table 1: Why did the authors not report DEEPEN-Adapt with Top-2 Ensembles? (DEEPEN-Adapt outperforms DEEPEN-Avg on all benchmark tasks with Top-4 Ensembles, so why is this not the default setting?)
- Figure 3: How do the baseline methods perform with an increasing number of ensemble members considered?
- Table 2 & 3: How do the baseline methods perform with the dense/sparse models and the generalist/specialist models?

**Limitations:**

The authors adequately addressed the limitations.

---

> ### Author Rebuttal · Authors · 2024-08-06
>
> We would like to thank you for your constructive feedback. We appreciate the opportunity to address your comments.
>
> &nbsp;
>
> **Question-1: Comparison between DeePEn with other ensemble methods in experiments beyond the main experiment.**
>
> Thanks for your insightful suggestion! We have followed your advice to supplement the results of the baseline ensemble methods (MinED, LLM-Blender, and Voting/MBR) in Fig.3  and Tab.2&3:
>
> **Figure-3** (Ensemble learning on various number of models):
>
> | Number of Models | Model Set     | Individual | LLM-Blender | MinED | Voting/MBR | DeePEn-Adapt (Ours) |
> | :--------------: | ------------- | :----------: | :-----------: | :-----: | :----------: | :-------------: |
> |1| LLaMA2-13B    |28.67|28.67|28.67|28.67|28.67|
> |2| +Mistral-7B |27.62|28.61| 28.45 |28.67| **30.65**|
> |3| +InternLM-20B |26.09|26.62|27.20| 30.06| **31.36**|
> |4| +Tigerbot-13B |22.71| 24.24| 29.50  | 30.28| **31.77**|
> |5| +Nanbeige-16B | 22.77| 22.63| 30.22 | 30.94| **31.02**|
> |6| +Skywork-13B  | 19.97| 22.71| 30.44 | 30.47| **31.16**|
> |7| +Yi-6B| 18.98| 21.25| 29.97 | 30.33| **30.50**|
>
> Please note that due to time limitations, we have only supplemented the results of baselines on NQ. The results on the other benchmarks (MMLU and PIQA) will be included in the next version.
>
> **Table-2** (Ensemble learning between the dense and sparse models):
>
> | Model                   | GSM8K             | PIQA              |
> | ----------------------- | ----------------- | ----------------- |
> | LLaMA2-70B (*Dense*)    | 63.84             | 71.27             |
> | Mixtral-8×7B (*Sparse*) | 65.73             | 71.88             |
> | LLM-Blender             | 64.52 (-1.21)     | 74.54 (+2.66)     |
> | MinED                   | 67.10 (+1.37)     | **75.65 (+3.77)** |
> | DeePEn (Ours)           | **67.33 (+1.60)** | 75.10 (+3.22)     |
>
> **Table-3** (Ensemble learning between the generalist/specialist models):
>
> | Model                     | En→De             | De→En             | En→Ro             | Ro→En             |
> | ------------------------- | ----------------- | ----------------- | ----------------- | ----------------- |
> | LLaMA2-13B (*Generalist*) | 30.60             | 42.27             | 30.83             | 39.99             |
> | NLLB-600M (*Specialist*)  | 32.30             | 41.49             | 31.91             | 42.39             |
> | LLM-Blender               | 33.26 (+0.96)     | 43.28 (+1.01)     | **33.17 (+1.26)** | 41.99 (-0.40)     |
> | MinED                     | 27.12 (-5.18)     | 36.83 (-5.44)     | 29.91 (-2.00)     | 34.39 (-8.00)     |
> | DeePEn (Ours)             | **33.34 (+1.04)** | **43.70 (+1.43)** | 32.95 (+1.04)     | **42.84 (+0.45)** |
>
> As the results show, unlike DeePEn, the **baseline ensemble methods perform quite unstably across various settings**. Specifically, MinED results in dramatic performance drops when ensembling the generalist LLaMA2-13B and the specialist NLLB-600B due to their large vocabulary divergence. LLM-Blender also leads to significant performance drops on NQ, GSM8K, and Ro→En translation due to its limited generalizability.
>
> We will add these results in our paper.
>
> &nbsp;
>
> **Question-2: Results of DeePEn-Adapt in Top-2 Ensembles?**
>
> Thanks for your important feedback! We have followed your advice to supplement the results of DeePEn-Adapt in the Top-2 ensembles of Tab.1:
>
> || MMLU| ARC-C| GSM8K| PIQA| TriviaQA| NQ|
> | ------------ | ----------------- | ----------------- | ----------------- | ----------------- | ----------------- | ----------------- |
> | LLM-Blender   | 63.85 (+0.60)     | 75.73 (-0.08)     | 54.89 (+0.99)     | 78.31 (+2.16)     | 74.10 (-0.22)     | 28.61 (-0.06)     |
> | MinED        | **65.04 (+1.79)** | 77.35 (+1.54)     | 18.50 (-35.40)    | 78.98 (+2.83)     | 72.30 (-2.02)     | 28.45 (-0.22)     |
> | DeePEn-Avg   | 64.68 (+1.43)     | **77.52 (+1.71)** | 55.42 (+1.52)     | 78.87 (+2.72)     | 75.90 (+1.58)     | 30.17 (+1.50)     |
> | DeePEn-Adapt | 64.41 (+1.16)     | **77.52 (+1.71)** | **55.65 (+1.75)** | **79.37 (+3.22)** | **76.08 (+1.76)** | **30.65 (+1.98)** |
>
> We will add these results to our manuscript.
>
> &nbsp;
>
> **Question-3: Why was the DeePEn-Adapt not the default setting in the top-2 ensembles?**
>
> Our original intention of devising DeePEn-Adapt was to alleviate the interference of lower-ranked models to the higher-ranked models. Therefore, we only tested DeePEn-Adapt in the top-4 model ensemble, where the 4th model could cause serious interference.
>
> Thanks for your valuable comment. In the future, we will set the DeePEn-Adapt as the default setting of DeePEn.
>
> &nbsp;
>
> **Question-4: What do the three colors (blue, yellow, purple) indicate in Fig.1?**
>
> Sorry for the unclarity. **Different colors indicate different clusters of samples**.
>
> To clearly illustrate that relative representations remain consistent across different models, we performed K-means to cluster the sampled representations and set different colors for different clusters.
>
> &nbsp;
>
> **Suggestion-1: Some passages could be condensed.**
>
> Thanks for your valuable feedback. **We will condense our manuscript in the future version.**
>
> In the original version of the paper,  the general paragraph of section 3.2 primarily describes the **implementation** of constructing relative transformation, including the "Anchor Selection" and "Normalization of relative representation matrix".
>
> In contrast, the specific paragraphs of  "Anchor Selection" and "Normalization of relative representation matrix" explain the **motivations** behind our implementations.
>
> &nbsp;
>
> **Suggestion-2:  Sections 3.3 and 3.5 both consider the aggregation of relative representations, currently separated by section 3.4.**
>
> Thanks for your valuable feedback. We will merge section 3.5 into 3.3.

---

> > ### Comment · Reviewer_c5c3 · 2024-08-08
> >
> > Thank you for the rebuttal. Your responses have adequately addressed my questions and concerns.

---

### Decision · Program_Chairs · 2024-09-25

**Decision:**

Accept (spotlight)

**Comment:**

The paper presents DEEPEN, an ensemble learning framework for heterogeneous LLMs that uses a novel training-free method to address vocabulary discrepancies across different models. This method translates probability distributions from individual LLMs into a shared “relative representation” space, facilitating aggregation and synthesis back into coherent output.

In addition to the innovative treatment of the token alignment problem inherent in multi-LLM integration, the proposed method performs well across multiple benchmarks. I agree with the reviewers that the strong theoretical grounding and the broad range of empirical validation are the strong points of this paper. The paper is also well-written and easy to understand without sacrificing details. The reviewers and I also appreciate the authors’ strong response in addressing the concerns around benchmark comparisons and methodological clarification during the rebuttal.

Even though I recommend acceptance for this paper, I recommend the authors pay attention to the weaknesses pointed out by the reviewers, especially regarding the depth of analysis concerning the generalization of DEEPEN across varied tasks and its sensitivity to hyperparameter settings.